# Variable rates of SARS-CoV-2 evolution in chronic infections

Ewan W. Smith[1], William L. Hamilton[2,3,4], Ben Warne[2,3], Elena R. Walker[1], Aminu S. Jahun[5], Myra Hosmillo[5], ISARIC Consortium[¶], Ravindra K. Gupta[2,6], Ian Goodfellow[5], Effrossyni Gkrania-Klotsas[3,7,8], M. Estée Török[2,3], Christopher J. R. Illingworth[1]*

1 MRC-University of Glasgow Centre for Virus Research, University of Glasgow, Glasgow, United Kingdom, 2 Department of Medicine, University of Cambridge, Cambridge, United Kingdom, 3 Cambridge University Hospitals NHS Foundation Trust, Cambridge, United Kingdom, 4 Wellcome Sanger Institute, Wellcome Trust Genome Campus, Hinxton, United Kingdom, 5 Division of Virology, Department of Virology, University of Cambridge, Cambridge, United Kingdom, 6 Cambridge Institute of Therapeutic Immunology & Infectious Disease (CITIID), University of Cambridge, Cambridge, United Kingdom, 7 MRC Epidemiology Unit, University of Cambridge, Level 3 Institute of Metabolic Science, Cambridge, United Kingdom, 8 School of Clinical Medicine, University of Cambridge, Cambridge, United Kingdom

¶ Membership of the ISARIC Consortium is provided in Supporting Information file S1 Text.
* christopher.illingworth@glasgow.ac.uk

## Abstract

An important feature of the evolution of the SARS-CoV-2 virus has been the emergence of highly mutated novel variants, which are characterised by the gain of multiple mutations relative to viruses circulating in the general global population. Cases of chronic viral infection have been suggested as an explanation for this phenomenon, whereby an extended period of infection, with an increased rate of evolution, creates viruses with substantial genetic novelty. However, measuring a rate of evolution during chronic infection is made more difficult by the potential existence of compartmentalisation in the viral population, whereby the viruses in a host form distinct subpopulations. We here describe and apply a novel statistical method to study within-host virus evolution, identifying the minimum number of subpopulations required to explain sequence data observed from cases of chronic infection, and inferring rates for within-host viral evolution. Across nine cases of chronic SARS-CoV-2 infection in hospitalised patients we find that non-trivial population structure is relatively common, with five cases showing evidence of more than one viral population evolving independently within the host. The detection of non-trivial population structure was more common in severely immunocompromised individuals (p = 0.04, Fisher's Exact Test). We find cases of within-host evolution proceeding significantly faster, and significantly slower, than that of the global SARS-CoV-2 population, and of cases in which viral subpopulations in the same host have statistically distinguishable rates of evolution. Non-trivial population structure was associated with high rates of within-host evolution that were systematically underestimated by a more standard inference method.

**Data availability statement:** The software package Blanche can be downloaded from the repository https://github.com/cjri/Blanche. IVY can be downloaded from the repository https://github.com/cjri/IVY. The latter repository includes anonymised data used in this project.

**Funding:** This work was supported by funding from the UK Medical Research Council (MC_UU_00034/1 and MC_UU_00034/6 and MC_ST_00034) provided to CI and ES and from the Wellcome Trust (207498/Z/17/Z and 206298/Z/17/Z) provided to IG. Data and Material provision was supported by grants from: the National Institute for Health Research (NIHR; award CO-CIN-01), the Medical Research Council (MRC; grant MC_PC_19059), and by the NIHR Health Protection Research Unit (HPRU) in Emerging and Zoonotic Infections at University of Liverpool in partnership with Public Health England (PHE), (award 200907), NIHR HPRU in Respiratory Infections at Imperial College London with PHE (award 200927), Liverpool Experimental Cancer Medicine Centre (grant C18616/A25153), NIHR Biomedical Research Centre at Imperial College London (award IS-BRC-1215-20013), and NIHR Clinical Research Network providing infrastructure support. The views expressed in this report are those of the authors and not necessarily those of ISARIC4C, NIHR, MRC, or PHE. The funders had no role in study design, data collection and analysis, decision to publish, or preparation of the manuscript

**Competing interests:** The authors have declared that no competing interests exist

## Author summary

Novel SARS-CoV-2 variants such as Alpha and Omicron were characterised by the presence of multiple genetic changes not found in other SARS-CoV-2 viruses at the time. One hypothesis is that these variants arose from cases of chronic SARS-CoV-2 infection, with the virus gaining variants during long-term infection before emerging back into the global viral population. A key parameter in assessing this hypothesis is the within-host rate of virus evolution. Measuring the within-host rate of evolution is made more complex if the virus is compartmentalised. If there are two or more independent viral populations within the host these may potentially evolve at different rates from one another. We here present the results of a novel statistical method to infer the presence or absence of independent viral populations in a host alongside their rates of evolution. Applied to nine cases of chronic SARS-CoV-2 infection, we find evidence for more than one independent viral population in five of the cases. Accounting for compartmentalisation leads us to infer more rapid viral evolution during chronic infection, supporting the hypothesis that chronic infections have an important role in global virus evolution.

## Introduction

The evolution of the SARS-CoV-2 virus during the early phases of the pandemic has been characterised by two apparently distinct processes [1]. In the first process, the viral population gained in diversity via the gradual accumulation of mutations. Mutations acquired in this process included those which conferred a fitness advantage upon the virus. For example, the D614G mutation in Spike protein was associated with higher viral load, and was observed at an increasing frequency over the course of 2020 within the UK [2]. The accumulation of mutations provided a distinct genetic signal, with the genetic distance of a virus from the original Wuhan strain increasing roughly linearly with time. In the second process, viral genomes were sometimes observed that differed by multiple mutations from anything else observed in the viral population. Notable examples of this were the Alpha and Omicron variants of concern [3,4]. These two processes have been understood as the combined result of within-host and between-host virus evolution. For example a viral population acquiring genetic substitutions during the course of a chronic case of infection might remain unobserved for a substantial period of time before spilling into the general population via a transmission event [5,6]. Multiple individual cases of chronic SARS-CoV-2 infection have been described in the literature [7–9]. In one example, the Δ69–70 deletion in the Spike protein, characteristic of the Alpha variant, was observed prior to the emergence of that variant during a case of chronic infection [10].

The hypothesis that chronic infection with SARS-CoV-2 lay behind the emergence of both the Alpha and Omicron strains imposes two demands upon the relevant cases of chronic infection. Firstly, to produce highly mutated viruses, evolution must

proceed faster during the period of chronic infection than it would in a pattern of regular cases of infection and transmission, generating and fixing multiple substitutions in the viral genome. Secondly, having acquired these mutations, the virus must transmit back into the regular population, maintaining sufficient fitness to compete with viruses that had not passed through a chronically infected host.

Positive selection, either for immune escape or to evade antiviral therapy, provides one mechanism via which rapid within-host evolution could occur. A study of multiple cases of SARS-CoV-2 evolution in immunocompromised hosts identified an increased proportion of substitutions in the Spike protein compared to the global viral population, supporting the hypothesis of positive selection [11]. Other analyses have identified increased burdens of mutation in viral populations from immunocompromised hosts [12,13], incidences of transmission of SARS-CoV-2 from cases of chronic infection [14], the emergence in immunocompromised hosts of variants conferring immune evasion [15], and the early emergence of variants yet to be found in the global viral population [16].

Although the existence of positive selection during SARS-CoV-2 infection is clear, the presence of selection does not imply an increased rate of virus evolution. Sequence data from selected cases of chronic SARS-CoV-2 infection have been used to identify more rapid rates of evolution during within-host infection than in the global population [8,17]. However, the overall picture is complex. An analysis of more than 100 cases of SARS-CoV-2 infection in hospitals did not find a generally increased rate of within-host evolution [18]. A recent overview of several hundred cases of chronic infection found examples of both apparently low and apparently high rates of within-host evolution [19].

Estimating within-host rates of virus evolution is a complex task, with multiple factors potentially confounding the calculation. Methods for phylogenetic analysis, used for evolutionary rate estimation in global RNA viral populations [20], assume that the sequences which make up the tree describe the genomes of individuals in a population [21]. When applied to a set of consensus sequences from a within-host viral population, this assumption is not correct. Phylogenetic methods usually do not account for errors in sequencing, which can increase terminal branch lengths [22]. Perhaps most significantly, efforts to infer within-host evolutionary rates rarely include a formal consideration of within-host population structure. Population structure has been hypothesised as contributing factor to the persistence of SARS-CoV-2 infection [23], with subpopulations potentially existing deep within the lungs [24] or elsewhere in the body [25]. For the purposes of rate estimation this complicates matters, in so far as distinct within-host populations might evolve at different rates [26]. A single rate of evolution, however inferred, may not reflect biological reality.

Here we use two novel approaches to gain an insight into within-host SARS-CoV-2 evolution. A first approach conducts sequence-based cartography, mapping consensus sequences, and therefore patterns of within-host evolution, in a simple visual manner. While only semi-quantitative in nature, this approach highlights potentially complex patterns of virus evolution, including potential substructure in the population, in cases of virus infection. A second approach performs a statistical calculation to infer the number of distinct viral populations within a host from what are assumed to be error-prone data, estimating rates of evolution for each population identified. Examining a set of 9 cases of chronic SARS-CoV-2 infection, we find evidence that within-host population structure, involving independently-evolving viral populations, is a common feature of infection. Subpopulations within the same host are found to evolve at significantly distinct rates, with a complex relationship pertaining between selection, viral phenotype, and the within-host rate of evolution.

## Results

Genome sequence data described cases of chronic infection in nine hospitalised individuals. Cases were selected under the criteria of having had viral genome sequences collected on at least four independent occasions, with these sequences spanning a period of infection of at least 21 days. Genome sequence data from one case (patient H) were described in an earlier publication [10]. The remaining cases originated from early in the pandemic, between March and June 2020. In this period, vaccination against SARS-CoV-2 had not been introduced [27], while specific antiviral treatments were few.

The application of a novel inference method to the sequence data provided statistical insight into within-host viral population structure during each infection. Our method fits models of different numbers of underlying populations to consensus viral sequence data from a host, assuming each sample to be representative of one of potentially multiple independent subpopulations of viruses in the host, and identifying the smallest number of subpopulations required to explain the sequence data. Our method infers a rate of evolution independently for each subpopulation.

Fig 1 shows different representations of consensus sequences collected from Patient G. Fig 1A shows an alignment of these sequences. Non-variant positions in the genome have been omitted; some sequences contained ambiguous nucleotides. Fig 1B shows a traditional phylogenetic reconstruction of these sequences, with samples coloured by the subpopulations to which they were allocated by our method. In the phylogeny, the samples on days 2 and 10 are clearly distinct from those on days 12, 22, 29, and 31, with other samples appearing identical to that from day 0. An alternative representation of the sequences, plotted using a method of dimension reduction, is shown in Fig 1C: Sequences have the appearance of evolving in different directions from each other.

Fig 1D conceptualises the way in which sequences are understood by our method. Assuming a simplified model of population dynamics, we divide sites in the viral genome into two, namely those at which a permanent change in the consensus genome sequence occurs, and those at which no such event occurs. Data in viral sequences is then treated as noisy observations of the underlying consensus. Observed sequence variation reflects either these 'fixation' events, or 'fluctuations', namely deviations from the true underlying state of the population. Our model fits a constant rate of evolution to a population, reflecting the number of fixation events: More fixation events in a given time implies a faster rate of evolution. By contrast, excessive fluctuation events are penalised. Biologically speaking, allele frequencies may fluctuate either side of the consensus cutoff, and genome sequencing may generate errors in the consensus, but these events occur at a limited rate. Allowing for more than one population in a host may allow fewer observed variants to be interpreted as fluctuations: Where this causes a significant improvement in model likelihood, the more complex population structure is inferred. In the case of Patient G, two subpopulations were identified. Subpopulation 2 was inferred to evolve faster than subpopulation 1, with three fixations in 10 days compared to four in 31 days. In our analysis no distinction could be made between the variants C5T and C7T. Where the first observed case of a variant is in the last collected sequence, the nature of this change is not easily recognisable. Under our maximum likelihood reconstruction one of C5T or C7T was inferred to be a fluctuation, with the other being a fixation, but with no ability to assign which was which. Full details are given in the Methods section.

We note that our method differs from some phylogenetic approaches in its treatment of ambiguous nucleotides. While ambiguous nucleotides do not contribute to our likelihood function, they are understood within the context of other samples, rather than being assigned as equal to the consensus. Contrary to the phylogenetic reconstruction of Fig 1B, the sequence collected on day 5 was inferred to have a G at positions 1 and 3, justified by observations of the same nucleotide on days 2 and 10, and the assignment of the sample to the same subpopulation. Representations for other populations are shown in S1 and S2 Figs.

Applied to simulated data our method showed a consistent ability to infer rates of viral evolution (S3–S6 Figs), while being conservative in the number of subpopulations identified. Although distinct subpopulations were not always identified from within a simulated host, overestimates of the number of subpopulations were rare, with one overestimate in 700 simulated populations.

Applied to our patient data, we inferred the presence of non-trivial population structure in five out of nine cases. In three of these cases two subpopulations were inferred to exist, while in one case three subpopulations were inferred. In patients G and H viral material was collected using different sample types (e.g., swab or sputum sample). No significant relationships were identified between sample type and the assignment of samples to subpopulations (S1–S3 Tables). Data from the remaining cases was consistent with a single underlying viral subpopulation.

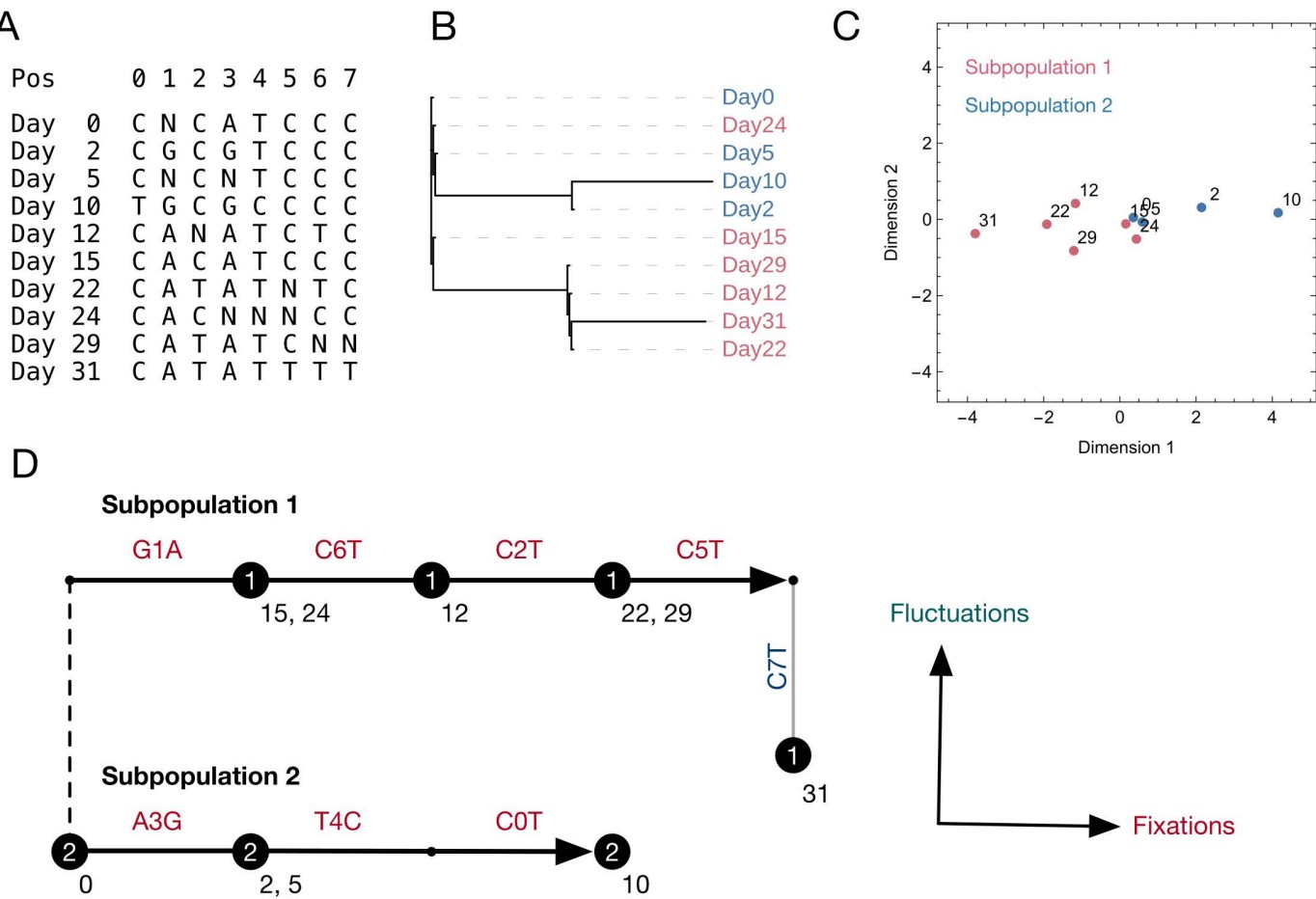

**Fig 1. Representations of sequence data from Patient G.** A. Alignment of sequences describing variant positions in sequences from different samples. Variant positions are renumbered from 0 to 7 for simplicity. B. Phylogenetic reconstruction of sequences, created using IQTree2[45]. Sequence labels have been coloured according to the subpopulations with which they were identified by our method. C. Graphical representation of sequences, calculated by a method of dimension reduction, whereby distances in the plot are fitted to match distances between sequences in sequence space. Subpopulations detected by our method are highlighted as red and blue dots. Numbers adjacent to dots describe the time of sample collection. These representations explicitly incorporate information from ambiguous nucleotides at variant sites, leading to the potential for points to be separated by distances of less than one unit. D. Interpretation of the maximum likelihood inference derived from our method. Samples are divided into two populations, which share a consensus sequence, represented by the vertical dashed line. Each subpopulation is inferred to gain sequence variants which eventually fix in the population; these are marked in red text. Other variants observed in the population are inferred to be temporary fluctuations in the sequence consensus which do not fix in the population; these are marked in blue text. Thick black horizontal arrows show the inferred evolution of each subpopulation. Collected sequences are shown as circles, are numbered by subpopulation, and have adjacent text marking the day of collection. Sequences are understood as stochastic observations of the underlying subpopulations. In the plot sequences are placed directly upon an arrow if all the variants they describe are fixations, or adjacent to the arrow if they describe both fixation and fluctuation events. We note that variants C5T and C7T cannot be distinguished from one another in our method; the locations of these variants within our plot could be interchanged.

Rates of evolution estimated by our approach showed a high degree of variability (Fig 2). Compared to estimated rates of evolution of $6 \times 10^{-4}$ substitutions per site per year for hospitalised SARS-CoV-2 patients [18], and $8 \times 10^{-4}$ substitutions per site per year for the global viral population [28], maximum likelihood rates from our model lay between zero and $3.9 \times 10^{-3}$ substitutions per site per year. In patients C and F we identified viral populations evolving significantly faster than the global rate, with a 95% confidence interval in the estimated rate for a subpopulation excluding the global rate of evolution. Conversely in patient A we identified a single population evolving significantly slower than the global rate. Where more than one subpopulation was identified in a host, estimated rates of evolution for the faster-evolving population were

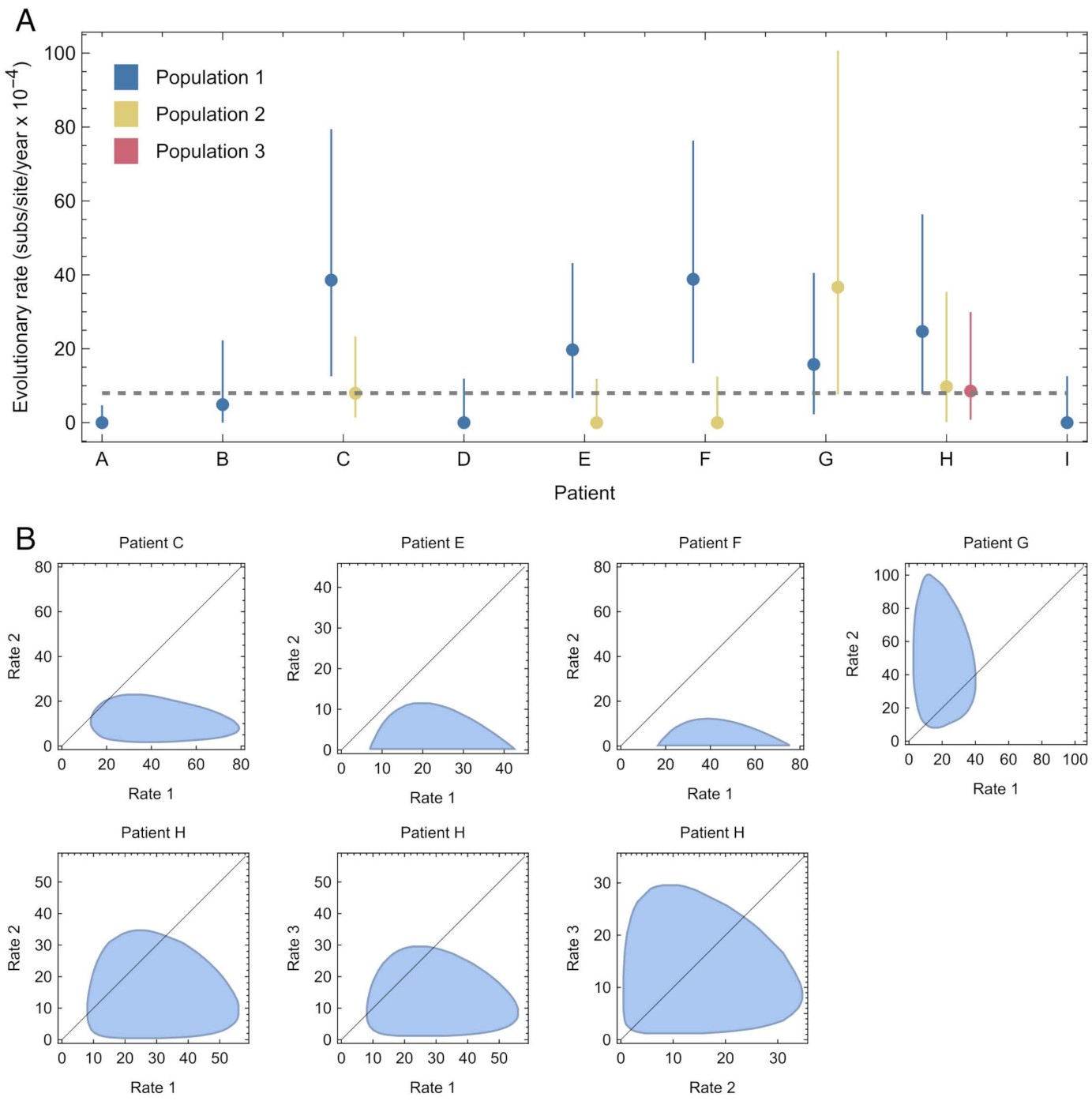

**Fig 2. Inferred rates of within-host SARS-CoV-2 evolution.** A. Maximum likelihood rates of within-host evolution are shown as dots, coloured according to the population identified within an individual. Between 1 and 3 populations were identified per host from the data. Vertical lines show estimated uncertainties in each rate. The horizontal gray dashed line shows an estimate for the global rate of SARS-CoV-2 evolution. B. Uncertainties in joint estimates of rates for individuals in which more than one population was identified. The black line shows parity between rates of evolution.

between 2.4 and 4.9 times that of the global population. In two of the five cases for which non-trivial population structure was identified, our method suggested that different subpopulations within the same host had significantly different rates of evolution (Fig 2B).

We further estimated rates of evolutionary change at synonymous and non-synonymous sites in the genome. Inferred rates of evolution at synonymous sites were generally faster than those at non-synonymous sites (Fig 3A). Among populations for which both synonymous and non-synonymous substitutions were observed, the combined rate of evolution was strongly correlated with the rate of gain of non-synonymous mutations (Fig 3B and S7 Fig). However, within these

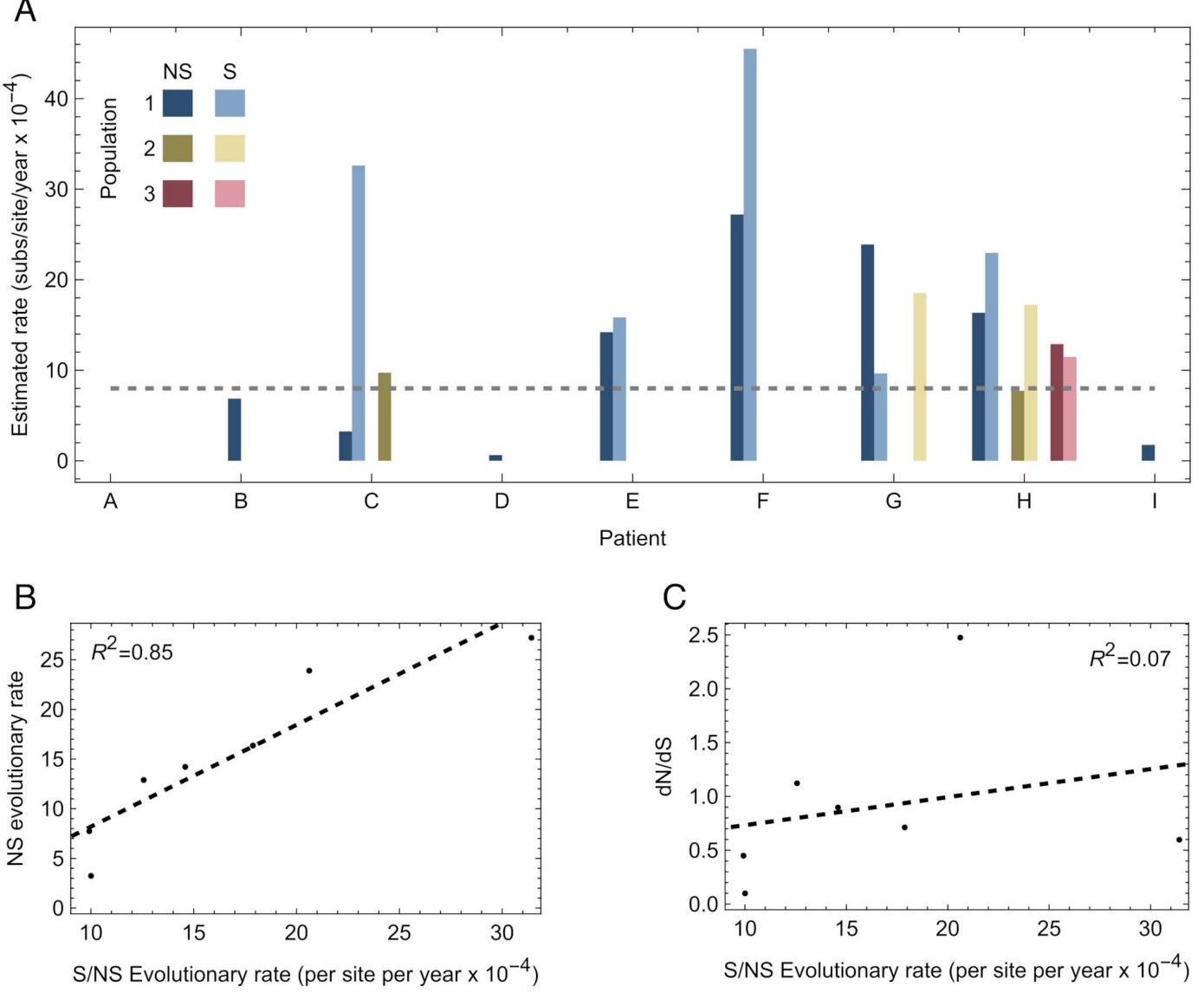

**Fig 3. Inferred rates of evolution at synonymous and non-synonymous sites.** A. Rates of evolution at synonymous and nonsynonymous sites were calculated across a statistical ensemble of model outputs. The horizontal dashed line shows an estimate of the global rate of SARS-CoV-2 evolution. B. Correlation between the inferred rate of evolution at nonsynonymous sites, and the total evolutionary rate calculated across both nonsynonymous and synonymous sites. The dashed black line shows a linear model fit to the data. C. Relationship between dN/dS and the total evolutionary rate calculated across both nonsynonymous and synonymous sites. The dashed black line shows a linear model fit to the data.

populations there was no significant link between a faster rate of evolution and a higher value of the statistic dN/dS (Fig 3C), as might arise if increased selection for non-synonymous change increased the rate of evolution. One challenge in measuring selection in this context is the small number of variants that arose in each individual; our individual-level dN/dS statistics may not be supported by enough data to identify underlying relationships between variant characteristics and evolutionary rate. Nonetheless, we did not find an association between the proportion of variants gained that were non-synonymous, and the rate of virus evolution.

Our data were consistent with a higher rate of substitutions in the Spike protein than in the remainder of the genome [11,19]. Due to the design of our model, nucleotide substitutions could be inferred to occur with probabilities strictly between zero and one. We identified an expected 9.1 substitutions in Spike, with an expected 45.2 substitutions in the rest of the genome. Together these suggest a substitution rate for Spike that was 36% higher than for the rest of the genome (Fig 4), although the difference was not great enough to be statistically significant. Indeed, the locations of inferred sites of nucleotide substitutions in the genome were consistent with a uniform distribution across the genome (p = 0.417, Kolmogorov Smirnov test). Roughly even distributions of variants have been noted for substitutions in the global SARS-CoV-2 population [11], and for SARS-CoV-2 in hamsters following treatment with mutagenic drugs [29].

Output from our analysis, which was based upon consensus sequence data only, was used to interpret short-read sequence data collected from patient H, described in an earlier study [10]. In the original analysis of this case, changes in allele frequencies showed rapid fluctuations in the frequencies of variants in the Spike protein. For example, viruses carrying the D796H mutation and a deletion of positions 69 and 70 (Δ69–70) rose to high frequency on two occasions shortly following the use of convalescent plasma therapy (Fig 5A). Our analysis of these data suggested that the observed fluctuations in allele frequencies could be broadly explained in terms of competition between stable subpopulations, with each subpopulation gaining distinct fixations in the Spike protein over time (Fig 5B). Rather than describing dramatic changes in a well-mixed population, the evolutionary dynamics of the system potentially reflect the dominance at different times of different subpopulations, either due to a selective response to variable antiviral therapy, or due to stochastic events in the emission and sampling of viral material from the host. We note that in this patient the D796H and Δ69–70 variants were associated with escape from convalescent plasma, while the Δ69–70 variant was one of the variants that defined the Alpha SARS-CoV-2 lineage which emerged in the global population some months after the clinical case [30]. Our decomposition identified both variants with a subpopulation in the host that was inferred to have a rate of evolution similar to that of the global population. As such, the gain of variants of experimentally-verified phenotypic importance was not associated

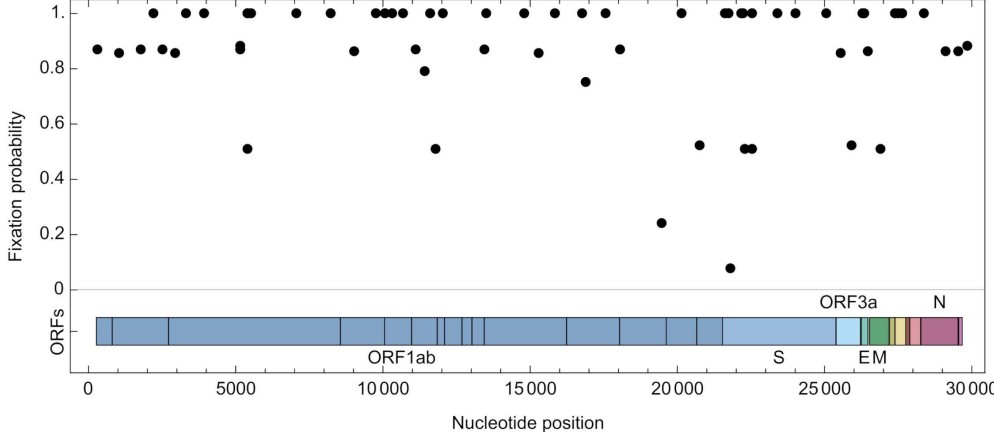

**Fig 4. Locations in the genome of potential fixation events.** Fixation events in our model are associated with a probability, which was calculated across an ensemble of models in which changes in the viral sequence could reflect either genuine change in a population or a form of sequencing error.

with an increased rate of virus evolution. The data of Fig 5 highlight limits in our method and potential future avenues for development. For example, the D796H variant, which appears at consensus level in samples assigned to subpopulation 2, appears as a minor variant in samples attributed to subpopulations 1 and 3 (Fig 5B), suggesting that viral material collected in a single sample may not arise solely from one underlying population. A method accounting for the detail of short-read viral sequence data would likely provide more information.

Our analysis of rates of evolution was repeated using a regression-based approach applied in other studies of within-host virus evolution [17,26]. This simpler approach, examining sequence divergence from the initial sample over time, did not capture the details of within-host population structure. For example, a regression calculation suggested a rate of evolution in patient F that was intermediate to the rates inferred for the different subpopulations we identified. For patients C and G regression suggested a rate of evolution that was consistent with one of the subpopulations identified by our approach, but not consistent with that of another subpopulation in the same patient (Fig 6 and S8 Fig). Of note, rates

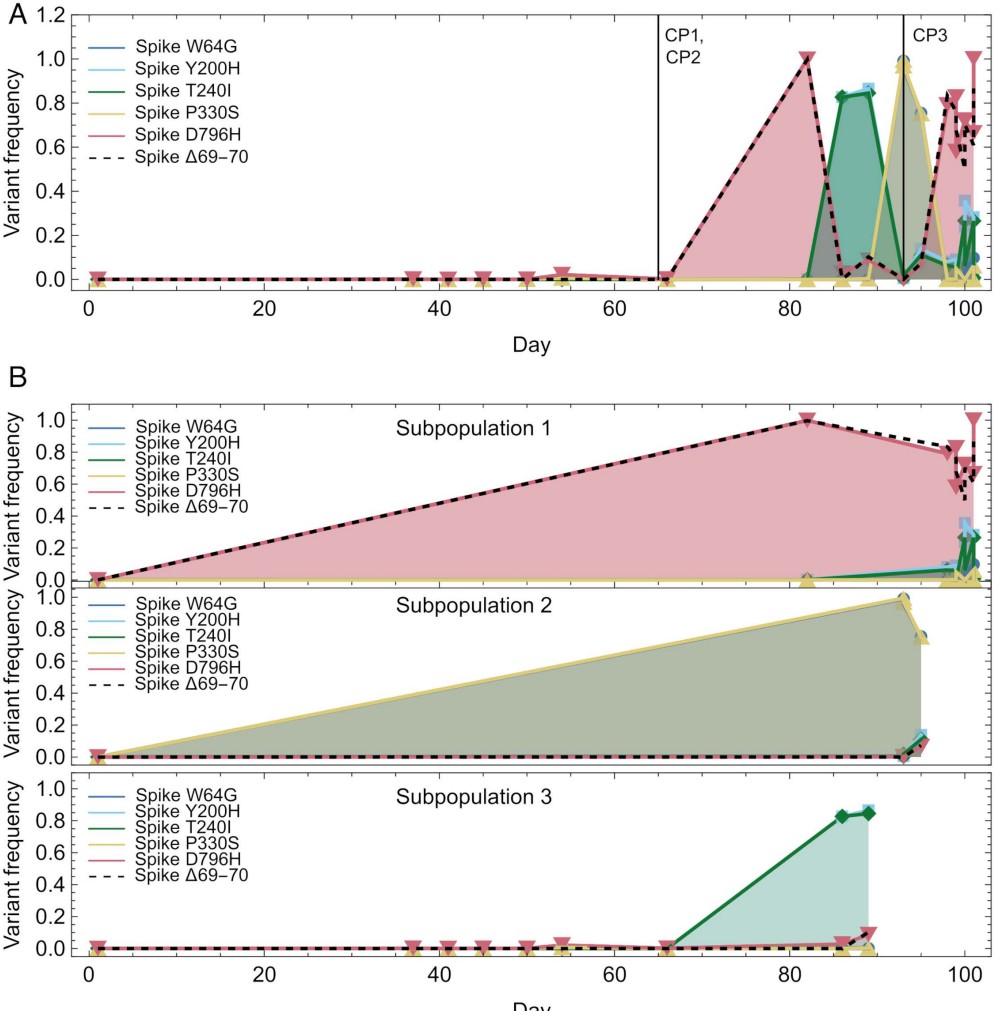

**Fig 5. Changes in allele frequencies in the data from patient H.** A. Allele frequencies of selected variants in the SARS-CoV-2 Spike protein, calculated from short-read sequence data from patient H. Vertical lines show the times of administration of three doses of convalescent plasma. B. Allele frequencies replotted according to a division of samples into subpopulations, as inferred by our method. Markers show the times of individual samples, with the first sample included in all subpopulations, representing the inferred initial consensus. Lines connecting variant frequencies are for illustration only.

estimated with the simpler regression model were universally lower than that of the faster-evolving subpopulation in each host where more than one subpopulation was identified; previous studies may have underestimated rates of within-host evolution through not accounting for population structure.

Clinical characteristics of the individuals in our study were explored but are not described in detail to preserve patient anonymity. Of the five patients in whom non-trivial population structure was identified, four were identified as being significantly immunocompromised, with conditions including primary immunodeficiency syndrome, haematological malignancy or organ transplantation. None of the remaining four patients was identified as being significantly immunocompromised. A previous study identified an increased rate of nucleotide substitutions in severely immunocompromised versus mildly or non-immunocompromised individuals [31]. In our dataset significantly immunocompromised individuals were more likely to have more complex population structure (p = 0.04, Fisher's exact test).

## Discussion

Within-host SARS-CoV-2 evolution may be more complex, and more varied, than has to date been appreciated. We have here applied a novel approach to study data from nine cases of chronic SARS-CoV-2 infection. Our method allows for a formal discrimination to be made between simple and complex population structures, inferring the presence of one or more independent viral subpopulations during a case of infection. Here, evidence for more than one subpopulation was found in five out of nine cases. We identified subpopulations evolving significantly faster, and significantly slower, than an estimated rate of evolution for the global SARS-CoV-2 population. However, rates of evolution were not linked in a simple manner to viral phenotype. For example, phenotypically validated mutations identified in patient H and associated with escape from convalescent plasma therapy did not arise in a subpopulation with a higher inferred rate of evolution.

Considered from a clinical perspective, we found a significant association between an individual being severely immunocompromised, and the identification of more complex within-host structure. Multiple potential explanations underlie this result. One possibility is that more severely immunocompromised individuals provide an environment that enhances the proliferation of multiple populations, through the colonisation by viruses of more regions within the airway. A second possibility is that the absence of host factors which would typically restrict the initiation of infection leads to higher transmission bottlenecks, with a more diverse viral population existing from the onset of infection. A third possibility is that our result has

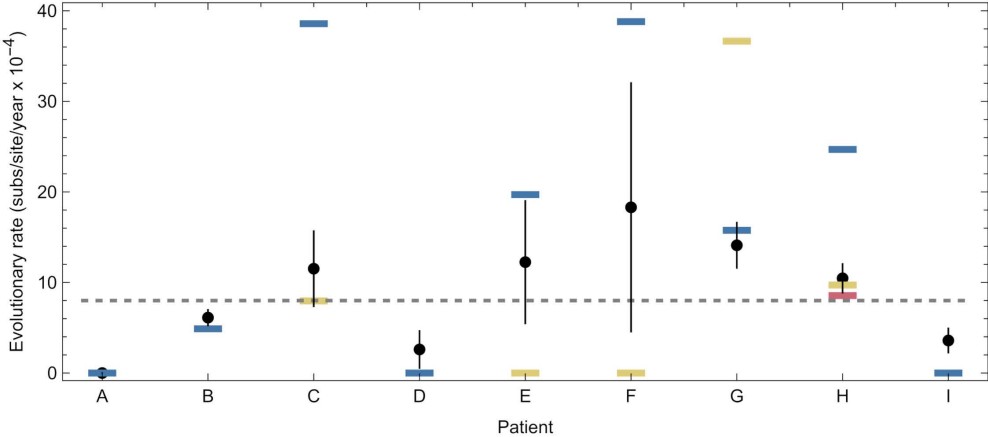

**Fig 6. Rates of evolution inferred using a simple method of linear regression.** The black dot for each patient shows an estimate of the within-host rate of virus evolution, inferred from a method of linear regression. Vertical black lines show error bars for these estimates. Blue, yellow, and red horizontal lines show the rates for the first, second, and third populations inferred by our approach. The linear regression method commonly underestimates the more rapid rates of evolution identified in structured cases of within-host evolution.

indirect cause. A longer period of infection in an immunocompromised host, or greater level of clinical monitoring of such an individual, could bias towards the collection of a greater number of samples, biasing our result towards the detection of complex structure even in the absence of a difference in structure itself.

Our study is not the first to identify subpopulations during within-host SARS-CoV-2 evolution, though our method has novelty in doing this in a formal statistical manner. In identifying distinct subpopulations, our method provides an advantage over simple linear-regression-based models. Distance metrics have in the past been used to identify sequences that may be associated with faster-evolving subpopulations [10]. Against these approaches, our method can identify different populations even where their rates of evolution cannot be distinguished. Indirect evidence, such as the partial onset of drug resistance, or a limited extent of interaction between distinct viruses in a host, has also been used to highlight potential within-host structure. While indicative, such data does not constitute formal statistical evidence [32,33]. Direct sampling from different anatomical locations in an infected individual provides the best evidence of spatial separation [34,35]. Viral proteins and RNA have been identified in multiple other organs within individuals, creating the potential for sequence diversity going beyond what was accessible to our study [36]. However, there are practical limitations in obtaining such samples in routine clinical care.

In common with other methods for inferring rates of evolution, our method is built around assumptions. For example, linear regression assumes that viruses in a host comprise a coherent population that evolves at a coherent rate. Our method relaxes the assumption of a single population but maintains that each population evolves at a constant underlying rate. In some cases, this assumption could affect the inference of the number of within-host populations. For example, in Patient E, sequences on days 12 and 21 equal to that on day 0 is followed by a sequence on day 31 containing five new mutations. In Patient F, something similar is observed, with sequences on days 20 and 22 differing by seven new mutations. In each of these cases, the assumption of a constant rate of evolution pushed the model towards a conclusion of emissions from differing populations, as opposed to a sudden gain of multiple variants. In particular, where antiviral drugs have the potential to accelerate evolutionary processes [37], a model of variable evolutionary rate may be appropriate. Models of time-dependent evolutionary processes have been used to study within-host evolution [38]. However, the limited nature of the data in this case mandated a cautious approach towards more complex evolutionary models. Among the individuals in our study, Patient H was repeatedly treated with convalescent plasma [10], suggesting that, while the division into subpopulations is strongly supported, a variable rate of evolution in each population might be considered. Within our data, conclusions in favour of distinct subpopulations are clearest where multiple samples contain shared, and distinct, variants. Inferred rates of evolution should be understood as mean values estimated over time. The timing of cases in our cohort, hospitalised in early-to-mid-2020, means that specific antiviral treatment was rare.

Other assumptions of our method may also influence the results. For example, the assumption that subpopulations are founded at the time of infection by a virus or viruses which share a common sequence reflects limited within-host sequence diversity during SARS-CoV-2 infection and a tight population bottleneck at the time of infection [39]. In the absence of higher numbers of samples it is difficult to distinguish subpopulations initiated by genetically distinct viruses from independently evolving populations founded by identical viruses. To the extent that our data show variants accumulating in populations over time, they support the hypothesis of virus evolution, as opposed to mere genetic diversity, present from the foundation of each infection. Next, our approach models fluctuations in the consensus sequences as occurring via a regular Poisson process. In this sense we do not explicitly account for cases in which individual sequences include large numbers of sequence errors. A different model of fluctuations could change the interpretation of data from Patients E and F. Thirdly, our model assumed that the detection of virus in quantities sufficient to produce sequence data implies the existence of a replicating viral population. If this assumption were not correct, the presence of residual viral genetic material could potentially explain apparently slower rates of evolution in some cases. We note that, if later samples from individuals are merely emissions from a dead virus population, the rates of evolution we report underestimate those for replicating virus populations. Finally, we assumed that each consensus sequence could be assigned

uniquely to a single subpopulation. This assumption is supported by some past data describing population structure in respiratory virus infection [10,26]. However, not all viruses follow this pattern, and it remains an assumption [40]. As we note for Patient H, some minor variants seemed to be shared between samples from distinct subpopulations, reflecting that our model, which is based upon consensus sequences, does not capture the full information provided by short-read data. Additional work is required to better understand the extent to which viral material collected via respiratory samples describes the viral population in a host.

Our results provide a novel perspective upon the impact of chronic infection upon the global SARS-CoV-2 viral population. Non-trivial population structure was associated with at least one subpopulation having an inferred rate of evolution substantially greater than that of the global population. Our detection of rapidly-evolving subpopulations within chronically infected hosts highlights the potential contribution of chronic cases to the evolution of the global SARS-CoV-2 population. The increased rates of evolution inferred by our method, arising from the account made for population structure, suggest that this potential may previously have been underestimated. Our study was unable to evaluate which of the variants generated during infection, if any, were transmitted to other individuals. However, it is interesting that the highest rates of evolution that we infer within-host (max. likelihood $3.9 \times 10^{-3}$ substitutions per site per year, upper bound $7.9 \times 10^{-3}$) are comparable with those inferred for branches leading to the emergence of novel SARS-CoV-2 variants (Beta $2.8 \times 10^{-3}$ per site per year; Alpha $8.5 \times 10^{-3}$ [41]). The rapid evolutionary processes generating novel variants may have occurred in compartmentalised populations. Further work is required to combine methods for studying within-host rates of evolution with larger and more descriptive datasets, to better understand the role of vaccination and antiviral therapies upon within-host evolution, and more broadly to understand the existence and consequences of virus compartmentalisation in immunocompromised hosts.

## Methods

### Ethics statement

This study was conducted as part of surveillance for COVID-19 infections under the auspices of Section 251 of the NHS Act 2006. As such, the collection of viral genome sequences and sample dates for this study did not require individual patient consent. The ISARIC/WHO Clinical Characterisation Protocol UK (ISARIC CCP-UK) was approved by the Oxford C Research Ethics Committee (reference: 13/SC/0149) and by the Scotland A Research Ethics Committee (reference 20/SS/0028). Symptom onset data provided by ISARIC CCP-UK was obtained by written consent. The COG-UK study protocol was approved by the Public Health England Research Ethics Governance Group (reference: R&D NR0195).

**Sequence data.** Our data comprised samples collected from nine patients who were treated for SARS-CoV-2 infection at Addenbrookes Hospital in Cambridge, UK. Identified positive samples were collected as part of a broader effort to investigate suspected hospital-acquired cases of infection [42]. In investigating suspected cases a multiplex PCR-based approach was used according to the modified ARTIC version 2 protocol with version 3 primer set, and amplicon libraries sequenced using MinION flow cells version 9.4.1 (Oxford Nanopore Technologies, Oxford, UK). Patients were selected from a dataset of cases of infection with samples collected prior to the end of October 2020, for whom i) at least four high-quality genome sequences were available and ii) the time interval between the collection of samples for the first and the last sequences was at least 21 days. Date of symptom onset was derived from the ISARIC study [43] for eight patients. Consensus level sequence data generated via nanopore sequencing were available for all nine patients [42]. Viral genome sequence coverage is shown in S9 Fig. To supplement these data we reanalysed short-read sequence data describing in more detail one of the nine cases of infection; a previous analysis of these data was published elsewhere [10]. Samples were collected via swab, sputum sample, or via aspirate. Diagnostics were carried out either using qPCR or using the Hologic Panther Fusion approach [44]. CT values for patients, where these were available, are shown in S10 Fig.

**Phylogenetic trees.** Phylogenetic trees were created for the purpose of visually illustrating relationships between viral genome sequences. Trees were conducted with iQTree2 [45] under default parameters. In each case the best fitting rate model was identified using the ModelFinder software package [46], which uses the Bayesian information criterion to fit a model of nucleotide substitution rates. Trees produced by this method were visualised using iToL [47]. Our method for identifying rates of evolution differs substantially from phylogenetic analysis: Where phylogenetics typically assumes that consensus sequences represent individuals drawn from an underlying population, we consider sequences as stochastic entities, representing noisy consensus samples from an underlying population (or one of multiple populations) in a host.

**Basic inference of subpopulations and rates of evolution.** Our inference method is based upon a framework of biological assumptions. Viral infection is founded in a host by a population of genetically similar viruses [26]. Viruses causing infection may comprise either a single coherent population of viruses, or more than one population, with populations being spatially segregated in the respiratory system, such that mixing between segregated populations is negligible [32], and each population evolves within a host in an independent manner. To the extent that selection acts upon viral populations, parallel adaptation, in the sense of specific mutations fixing in more than one independent populations, is rare [48]. We note that the majority of cases in this study were not treated with specific antiviral therapies, due to their occurrence early in the pandemic.

We inferred rates of evolution using models of viral populations which included successively higher numbers of distinct subpopulations, denoted I, V, Y, and X. We used model selection (i.e., the Bayesian Information Criterion [49]) to identify the model which best explained the data.

**Conversion of sequence data to binary code.** Variant positions in consensus viral sequences were identified. Assuming no more than two alleles per site, the aligned consensus sequences were converted into binary code, with a 0 representing the first observed allele at a site in the genome and a 1 representing the variant allele. Ambiguous nucleotides were marked as an N, while non-variant sites were removed from consideration. To these data we added an initial consensus sequence at time zero, by default equal to the sequence of the first sample observed within the patient. Further calculations were conducted using these binary data (Fig 7A and 7B).

**Allocation of samples to subpopulations.** Samples were divided into subpopulations, with each sample uniquely ascribed to one of between one and four subpopulations (Fig 7C). Each population shared the initial consensus sequence. Constraints were placed upon the division of samples. Firstly, at least one subpopulation had to be represented by three or more samples. Next, we made the assumption that different subpopulations within the host evolve independently from one another. Noting the large size of the genome relative to the number of variant sites, we then favoured explanations in which variants did not arise multiple times in independent populations. Where a variant was observed in more than one subpopulation we applied a penalty of log v/L to the model likelihood for each such occurrence, where v was the number of variant sites, and L was the total length of the viral genome. Further, we excluded any division of samples whereby two sequences in different populations shared two or more variants. This exclusion had a pragmatic purpose, greatly reducing the number of ways of dividing samples between subpopulations, and facilitating more rapid calculations.

**Fixation sites.** Having defined subpopulations, variant positions were re-calculated, identifying 'fixation' sites for each subpopulation (Fig 7D), representing sites in the genome for which the modelled underlying consensus sequence underwent a permanent change. Across the samples representing a subpopulation, we calculated the Hamming distance between the binary vector representing the presence or absence of a variant at each locus to two sets of vectors. The set $V_0$ comprised two vectors, one consisting of all zeros, and the other consisting of all ones. The set $V_1$ comprised the set of 'fixation' vectors, with a string of zeros followed by a string of ones (S11 Fig). Where the minimum Hamming distance to a vector in set $V_1$ was smaller than the minimum Hamming distance between to a vector in set $V_0$, a variant position was classified as a fixation.

**Fixation times.** Times of fixations were inferred as the interval between the first observation of a 1 in at a variant site, and the previous observation from that subpopulation (Fig 7D). For any given time interval between samples, we

**A**

```
          0 1 2 3 4 5 6 7
Day Sequence
      0 1 2 3 4 5 6 7
  0   C N C A T C C C
  2   C G C G T C C C
  5   C N C N T C C C
 10   T G C G C C C C
 12   C A N A T C T C
 15   C A C A T C C C
 22   C A T A T N T C
 24   C A C N N N C C
 29   C A T A T C N N
 31   C A T A T T T T
```

**B**

```
Day Binary data
Pos 0 1 2 3 4 5 6 7

 0C 0 0 0 0 0 0 0 0

  0   0 N 0 0 0 0 0 0
  2   0 0 0 1 0 0 0 0
  5   0 N 0 N 0 0 0 0
 10   1 0 0 1 1 0 0 0
 12   0 1 N 0 0 0 1 0
 15   0 1 0 0 0 0 0 0
 22   0 1 1 0 0 N 1 0
 24   0 1 0 N N N 0 0
 29   0 1 1 0 0 0 N N
 31   0 1 1 0 0 1 1 1
```

**C**

```
Subpopulation 1                Subpopulation 2

Day Binary data                Day Binary data
Pos 0 1 2 3 4 5 6 7            Pos 0 1 2 3 4 5 6 7

 0C 0 0 0 0 0 0 0 0             0C 0 0 0 0 0 0 0 0

 12  0 1 N 0 0 0 1 0             0   0 0 0 0 0 0 0 0
 15  0 1 0 0 0 0 0 0             2   0 0 0 1 0 0 0 0
 22  0 1 1 0 0 N 1 0             5   0 N 0 N 0 0 0 0
 24  0 1 0 N N N 0 0            10   1 0 0 1 1 0 0 0
 29  0 1 1 0 0 0 N N
 31  0 1 1 0 0 1 1 1
```

**D**

```
Subpopulation 1                                      Subpopulation 2

Day Binary data  Fix  Fluc  Fixation   Day Binary data  Fix  Fluc
Pos 1 2 5 6 7                times      Pos 0 3 4

 0C 0 0 0 0 0                             0C 0 0 0

 12  1 N 0 1 0     2    0      ← 1 →       0   0 0 0
 15  1 0 0 0 0     0    0      ← 2 →       2   0 1 0       1     0
 22  1 1 N 1 0     1    0      ← 3 →       5   0 N 0       0     0
 24  1 0 N 0 0     0    0      ← 4 →      10   1 1 1       2*    0
 29  1 1 0 N N     0    0      ← 5
 31  1 1 1 1 1     2*   0      ← 6        Fix  1 1 1

Fix 1 1 1 1 1                             T   4 2 4

  T  1 3 6 3 6
```

**Fig 7. Processing of sequence data.** A. Viral genome sequences were reduced to the set of loci at which variants were found. Ambiguous nucleotides are represented by an N. Data shown are from Patient G. B. This alignment was converted into binary code, representing consensus and variant alleles. The sequence 0C represents the consensus. Variants in the genome are labelled by position, from 0 to 7. C. Sequences were split into subpopulations, each with the same consensus. One example splitting is shown, representing the optimal split for these data. Multiple alternative splittings are possible. D. Non-variant sites were removed for each subpopulation. Fixations were identified for each subpopulation, being shown via a 1 in the 'Fix' row below the sequence data. Timings of each fixation are shown as the interval in which they occurred (red text). For each sample after the proposed consensus, the numbers of fixations and fluctuations in that sample are shown. Fixation and fluctuation numbers, and the respective days of their observation, were used to infer rates of evolution for each subpopulation. Fixation numbers for events occurring in the final time-point are starred; these were modelled as potentially describing either fixations or fluctuations in the sequence, calculating all possible likelihoods according to Equation 4.

calculated the number of fixations occurring in that interval. The number of fluctuations in a sample was defined as the number of observations of a 1 that were not at fixation sites, plus the number of observations of a 0 at fixation sites after the time of a fixation event.

**Likelihood calculation.** Fixation and fluctuation numbers were used to calculate the maximum likelihood rate of evolution for each population. We specified a rate of evolution $\lambda$, measured in units of number of fixations per day, and an 'error' parameter $\varepsilon$, measured in units of the number of variants per sequence. The likelihood of these two rates was described by

$$L(\lambda, \in) = \sum_{i=2}^{k} \log \left[ \text{Poisson} \left( n_i^{fix}, \lambda \Delta_{t_i} \right) \text{Poisson} \left( n_i^{fluc}, \in \right) \right]. \tag{1}$$

Where $n_i^{fix}$ is the number of fixations in the time period between the collection of sample $i$ and the previous time-point, $\Delta_{t_i}$ is the time in days between the collection of sample $i$ and the previous time-point, $n_i^{fluc}$ is the number of fluctuations in sample $i$, and

$$\text{Poisson}(n, r) = \frac{r^n e^{-r}}{n!} \qquad (2)$$

Fluctuations in the first time point (i = 1, above) were not considered, as this was defined by the initial consensus. We note that fluctuations in the consensus sequence could be caused by multiple factors, including formal errors in sequencing, but also clonal competition between variants. We retain the 'error' terminology used in a previous publication [18] while remaining ambiguous to the true cause of sequence fluctuations.

The log likelihood of a model was calculated as the sum of the log likelihoods $L(\lambda, \varepsilon)$ across all subpopulations. An improvement in the Bayesian Information Criterion of 10 units was required for the acceptance of a more complex model, representing a conservative cutoff for the acceptance of additional subpopulations. In our model we inferred independent values of $\lambda$ for each subpopulation. The value of $\varepsilon$ was fixed to a constant value of 0.207 nucleotide errors per sequence, using a value inferred from a large set of sequences from 136 patients at Addenbrookes Hospital, sequenced using the same protocol [18]. Investigation of those data did not identify a strong effect of CT value upon the number of nucleotide errors in a sequence [18].

**Initial and final status of subpopulations.** Our model presumes that each subpopulation has a common initial consensus sequence, and evolves to some final state in sequence space. Defining the initial and final states of the system involved further calculations.

**Final state of each subpopulation.** In the model described, the observation of novel variants in the final sequence from a subpopulation creates some ambiguity, as these variants could represent either fixations in the population or simply fluctuations arising from clonal competition or sequencing error. These events were treated in a flexible manner. If for a population there were $n_k^{fix}$ fixations in the final time interval, we considered the ensemble of cases for which there were s fixations, and $n_k^{fix}$-s additional fluctuations observed in the final time point, for different integer values of s between 0 and $n_k^{fix}$. The maximum likelihood model was then defined as the maximum likelihood calculated for any case within this ensemble.

$$L(\lambda, \in) = \max_{s \in \{0, n_k^{fix}\}} \left\{ L\left(\lambda, \in | s\right) \right\} \qquad (3)$$

**Defining the initial consensus.** Our approach assumed that any subpopulations within a host were founded from a common initial consensus sequence, rather than by viruses with distinct sequences. The initial consensus sequence was defined in a model-specific way, summarised in S12 Fig, as follows:

**Initial consensus: Model I: One subpopulation.** Where data were available, the time of symptom onset for a patient was denoted as day zero; in the absence of data the first collected sample was denoted as day zero. If the first available sample was collected on day zero of infection, this sample was used as the initial consensus; evolution was modelled from this point forwards. If the first sample was not collected on day zero, the potential exists for substitutions to have occurred in the viral population prior to the collection of the first sample. We denote by $n_{pre}$ the number of fixations prior to the collection of the first sample, and optimised over this parameter.

$$L(\lambda, \in) = \max_{n_{pre}, s \in \{0, n_k^{fix}\}} \left\{ L\left(\lambda, \in | s, n_{pre}\right) \right\}. \qquad (4)$$

**Initial consensus: Model V: Two subpopulations.** Model V describes the evolution of two distinct viral populations, which are separated at the moment of initiating infection. Within this model we have two subpopulations, each of which has a first-collected sequence; these two sequences may differ by some number of variants. An ensemble of models were generated in which the consensus sequence was one of the set of possible sequences including the two first-observed sequences and any possible sequence intermediate to the two (S12 Fig). The maximum likelihood was then calculated, considering all possible consensus sequences in turn.

**Initial consensus: Models Y and X: Three or four subpopulations.** Models Y and X describe the evolution of three and four populations respectively, with a common initial consensus. In constructing a consensus, where the first sample from only one population described a variant, that variant was assumed to have arisen in that population. However, sites for which the first samples from more than one population contain a variant are ambiguous. An ensemble of models were generated in which the consensus sequence was one of the set of possible sequences with a 0 at every site for which a variant was seen in the first samples from zero or one populations, and either a 0 or a 1 at every site for which a variant was seen in the first samples from two or more populations. The maximum likelihood was then calculated, considering all possible consensus sequences in turn.

**Likelihood calculations and fixation probabilities.** Maximum likelihood values of rate parameters were calculated using a simple optimisation routine, considering all possible assignments of samples to subpopulations. Upper (and lower) bounds for single rate parameters were conducted using a ratcheting process whereby a parameter was fixed to only increase (or decrease) while other parameters were free to change, subject to the likelihood being within two units of the maximum likelihood. Where different initial sequences or values of the parameter $s$ led to likelihoods within two units of the maximum likelihood, calculations were repeated across all such initial sequences and values, finding extreme values subject to the likelihood constraint. The identification of plausible combinations of rate parameters, used to generate Fig 2B, was conducted on a grid, altering parameters by units of 0.002, identifying parameter combinations that produced likelihoods within 2 units of the maximum.

In our model the fixation of a variant can occur in some, but not in all, scenarios. We used the likelihood of each scenario to calculate a probability of fixation, equal to the weighted sum over all parameter choices of a binary value denoting fixation, weighted by the respective scenario likelihoods.

$$P\left(\text{fix}\right) = \frac{\sum_{s,a,c} 1_{fix} e^{L(\lambda, \in |s,c)}}{\sum_{s,a,c} e^{L(\lambda, \in |s,c)}}.$$

(5)

where the sum was calculated over all possible assignments of samples to populations $a$, all possible starting sequences, $c$, and all possible decisions about fixations in the last time point $s$. The value $1_{fix}$ here equals one if the variant fixed given $s$, $a$, and $c$, and zero otherwise.

Rates of substitutions at synonymous and non-synonymous sites were calculated in terms of the number of fixation events at each type of site per day of evolution. Suppose that $P(\text{fix})_{i,j,k}$ denotes the probability that variant $k$ fixes in subpopulation $j$ in patient $i$. Then the rate of non-synonymous evolution for this population was calculated, in units of substitutions per nucleotide per day, as

$$R_{i,j}^{NS} = \frac{\sum_{\{k|k \ is \ NS\}} P(\text{fix})_{i,j,k}}{T_i o_i^{NS}}$$

(6)

Where o denotes the number of sites in the genome that would provide non-synonymous mutations, and $T_i$ is the length of time over which samples were collected from population $j$ in individual $i$. The rate of evolution at synonymous sites was calculated in a similar manner.

**Mapping consensus viral sequences.** We developed the Blanche software package to perform sequence-based cartography, inspired by the use of similar methods for plotting genetic and antigenic data [50,51]. Blanche carries out a simple dimension reduction calculation to facilitate the plotting of viral genome sequence data collected from individual patients. Given a set of viral sequences, we first calculated a matrix $D^H$ of Hamming distances between sequences. Ambiguous nucleotides in the data were represented in our calculation by their expected value. Where a sequence contained an ambiguous nucleotide (e.g., N) at a variant site in the population, this nucleotide was set to the expected value given the composition of other sequences from the same individual. For example, if at a site there were three

unambiguous observations of an allele A, and two observations of an allele C, an ambiguous nucleotide would be defined as (0.6 A, 0.4C). The distance from this nucleotide to an A would then be equal to 0.4, while the distance from this nucleotide to a C would be 0.6.

Given values for the matrix $D^H$, we then used a simple optimisation process to identify a set of points in two-dimensional Euclidean space, with matrix of Euclidean distances between points $D^E$, so as to minimise the sum of the squared distances between matrix elements

$$D = \sum_{i,j} \left( D_{i,j}^E - D_{i,j}^H \right)^2$$
(7)

In this calculation two-dimensional Euclidean space was chosen to provide a simple representation of points on a page or screen. We note that for many datasets the minimum value of $D$ was non-zero: Our representations provide an approximate picture of the relationships between genome sequences.

**Simulated data.** We generated simulations describing the accumulation of mutations in simple and mixed populations. Simulations were analysed using IVY, making use of the GNU parallel software package [52]. In a simple population, evolution was modelled as occurring via a simple process of the gain of mutations. The initial sequence of the population was specified, somewhat arbitrarily, as the Wuhan Hu-1 strain of the SARS-CoV-2 virus. The population was then modelled in terms of a consensus sequence.

Sequences were sampled from our population at regular times, spaced at intervals $\Delta t$. At each sampling point, the viral sequence gained a number of consensus mutations defined as the outcome of a Poisson process with rate $\lambda$. Specifically, the number of changes to the sequence was calculated as

$$n_s \sim \text{Poisson}(\lambda \Delta t)$$
(8)

Changes in the sequence were modelled to occur at uniformly randomly sampled positions within the sequence. No distinction was made between changes that caused synonymous, non-synonymous, or nonsense mutations.

Further to the changes in the viral population, we modelled errors as arising in the sequence at rate $\mu$, independent of time, so that the observed sequence contained $n_e$ errors, where

$$n_e \sim \text{Poisson}(\mu)$$
(9)

Errors also occurred at uniform random positions in the sequence.

By default we collected samples at seven time points, with the first sample collected at time zero, and $\Delta t = 5$ days. Simulations were calculated for values of $\lambda$ in {0.1, 0.2, 0.3} day$^{-1}$ and with $\mu = 0.206845$.

Mixed populations were generated in a similar manner to simple populations, but with two or three underlying viral populations that shared an initial consensus sequence. Each population evolved independently of the other, with an independent rate of evolution, again in {0.1, 0.2, 0.3} day$^{-1}$. At each sampling point the consensus sequence of one of the underlying populations was recorded, subject to error. The schedule of which population a consensus was collected from at each sampling point was randomly chosen to have an equal probability of sampling either population at a given time point, but with the restriction that over time, a total of at least two samples were collected from each population. Samples from simulations involving three underlying populations were collected at ten five-day intervals, with the final sample collected on day 45.

## Supporting information

**S1 Fig.: Representations of viral sequence data.** For each patient not so described in the main text, graphical representations of sequence data are shown. Dots are labelled with the day of the collection of the sample they represent and

are coloured by the subpopulation with which they were identified by our method. A phylogenetic representation of the sequence data is shown below each of the respective plots.
(PDF)

**S2 Fig.: Further representations of viral sequence data.** Representations of data show the samples attributed to each population (black, numbered by subpopulation, positioned by sequence). Nucleotide substitutions for each population are indexed from zero. Fixations within subpopulations are shown in red text, while fluctuations are shown in blue text. Maximum likelihood reconstructions are shown; there are often multiple plausible reconstructions of the data.
(PDF)

**S3 Fig.: Inferred rates of evolution from simulated data.** Simulations described simple populations with an underlying rate of evolution, simulated for 30 days. The actual number of mutations gained by a population during this period is Poisson distributed according to the product of the rate and time. **A.** Inferred rates of evolution (gray bars) given a rate of 0.1 per day. The mean inferred rate across 100 populations (vertical blue line) is very similar to the actual rate (vertical red line). **B.** Inferred 95% confidence intervals for these inferences. The correct rate of evolution (vertical red line) was contained within 93 of 100 intervals. **C.** Inferred rates of evolution given a rate of 0.2 per day. The mean inferred rate (vertical blue line) is very similar to the actual rate (vertical red line). **D.** Inferred 95% confidence intervals for these inferences. The correct rate of evolution (vertical red line) was contained within 98 of 100 intervals. **E.** Inferred rates of evolution given a rate of 0.3 per day (gray bars). The mean inferred rate (vertical blue line) is very similar to the actual rate (vertical red line). **F.** Inferred 95% confidence intervals for these inferences. The correct rate of evolution (vertical red line) was contained within 98 of 100 intervals.
(PDF)

**S4 Fig.: Inferred rates of evolution from simulated data.** Data describe populations with two subpopulations, simulated for 30 days. **A.** Inferred rates of evolution (gray bars) where viral populations evolve at rate 0.1. The mean inferred rate (vertical blue line) and actual rate (vertical red line) are shown. **B.** Inferred rates of evolution (gray bars) where viral populations evolve at rates 0.1 and 0.2. The mean inferred rate (vertical blue line) and actual rate (vertical red line) for each population are shown. **C.** Inferred rates of evolution (gray bars) where viral populations evolve at rates 0.1 and 0.3. The mean inferred rate (vertical blue line) and actual rate (vertical red line) for each population are shown.
(PDF)

**S5 Fig.: Numbers of populations inferred from simulated data.** Data describe populations with one or two subpopulations, simulated for 30 days. Under-calling of distinct populations occurs in up to 25% of inferences for the case where the rates of evolution of the populations are identical, but over-calling of distinct populations was rare.
(PDF)

**S6 Fig.: Statistics from applications to simulated data with three subpopulations.** A. Number of subpopulations identified by the model. Three subpopulations were identified in 97 out of 100 simulations. **B.** Smallest fitness parameter inferred by the model shown as a histogram (gray bars). The red line and blue line show the simulated rate and the mean inferred rate of evolution respectively. **C.** Intermediate fitness parameter inferred by the model shown as a histogram (gray bars). The red line and blue line show the simulated rate and the mean inferred rate of evolution respectively. **D.** Largest fitness parameter inferred by the model shown as a histogram (gray bars). The red line and blue line show the simulated rate and the mean inferred rate of evolution.
(PDF)

**S7 Fig. Correlation between the inferred rate of evolution at nonsynonymous sites, and the total evolutionary rate, including non-coding sites.** The dashed black line shows a linear model fit to the data.
(PDF)

**S8 Fig. Rates of evolution inferred using a simple method of linear regression.** Individual sequences are shown as dots. Non-integer sequence distances were sometimes identified in the case of missing nucleotide data. The black dashed line for each case shows a linear regression model fitted to the data. The gray shaded region shows confidence intervals for this model.
(PDF)

**S9 Fig. Extent of coverage for viral genome sequencing.** Coverage is described as the fraction of the genome for which unambiguous nucleotides were observed.
(PDF)

**S10 Fig. CT values for samples collected from patients, where recorded.** Times are shown relative to the time of symptom onset, where known, or to the time of the first collected sample. Samples from Patient I were generated using the Panther platform, such that no CT values were recorded.
(PDF)

**S11 Fig. Examples of sets $V_0$ and $V_1$.** These sets of vectors are used in the separation of fixation and fluctuation events. $V_0$ represents the continual presence or absence of a variant in the population, while $V_1$ represents the fixation of a variant at some point in time.
(PDF)

**S12 Fig. Determination of consensus sequences.** Consensus sequences were defined in a flexible manner, under the assumption that fixations, once gained, are not lost. In model I, with one subpopulation, where the first sample is collected at time $t > 0$, the potential exists for fixations to have occurred in the population prior to time t. Our model allows for some number $n_{pre}$ fixations to have occurred. In model V, with two subpopulations, where the first observed sequences in each subpopulation differ, we allow the consensus to have been either of these two sequences, or any sequence in between them. In models Y and above, with three+ subpopulations, the initial consensus is defined uniquely as not containing any of the variants observed in the first sequences of the distinct populations. Where multiple consensus sequences were possible, the maximum likelihood was calculated across all possible such sequences.
(PDF)

**S1 Table. Counts of swabs and sputum samples from patient G, alongside the subpopulations to which they were assigned.** No significant association was identified between subpopulation and type of sample (p-value 0.21, Fisher exact test). Samples of unknown sampling method (1 sample) were neglected from the analysis.
(XLSX)

**S2 Table. Counts of swabs and sputum samples from patient H, alongside the subpopulations to which they were assigned.** No significant association was identified between subpopulation and type of sample (p-value 0.28, Fisher exact test). Aspirate samples (1 sample) were neglected from the analysis.
(XLSX)

**S3 Table. Counts of swabs and sputum samples from patient H, alongside the subpopulations to which they were assigned.** No significant association was identified between subpopulation and type of sample (p-value 0.53, Fisher exact test). Aspirate samples (1 sample) were neglected from the analysis.
(XLSX)

**S1 Text. Membership of the ISARIC Consortium.**
(DOCX)

## Acknowledgments

We thank Iain Barrass for research computing support. This work uses Data/ Material provided by patients and collected by the NHS as part of their care and support #DataSavesLives. The Data/ Material used for this research were obtained from ISARIC4C. The COVID-19 Clinical Information Network (CO-CIN) data was collated by ISARIC4C Investigators.

## Author contributions

**Conceptualization:** Chris Illingworth.

**Data curation:** William L Hamilton, Ben Warne, Effrossyni Gkrania-Klotsas, Chris Illingworth.

**Formal analysis:** Ewan W Smith, Elena R Walker, Chris Illingworth.

**Funding acquisition:** Estee M Török, Chris Illingworth.

**Investigation:** Ewan W Smith, Elena R Walker, Aminu S Jahun, Myra Hosmillo, Chris Illingworth.

**Methodology:** Chris Illingworth.

**Project administration:** Ian Goodfellow, Effrossyni Gkrania-Klotsas, Estee M Török, Chris Illingworth.

**Resources:** William L Hamilton, Ben Warne, Aminu S Jahun, Myra Hosmillo, The ISARIC Consortium, Ravindra K Gupta, Ian Goodfellow, Effrossyni Gkrania-Klotsas, Estee M Török.

**Software:** Chris Illingworth.

**Supervision:** Ian Goodfellow, Chris Illingworth.

**Validation:** Chris Illingworth.

**Visualization:** Chris Illingworth.

**Writing – original draft:** William L Hamilton, Chris Illingworth.

**Writing – review & editing:** Ewan W Smith, William L Hamilton, Ben Warne, Elena R Walker, Aminu S Jahun, Myra Hosmillo, The ISARIC Consortium, Ravindra K Gupta, Ian Goodfellow, Effrossyni Gkrania-Klotsas, Estee M Török, Chris Illingworth.

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
