## [Decision Letter · Decision Letter 0]

28 Aug 2024

Dear Dr Illingworth,

Thank you very much for submitting your manuscript "Variable rates of SARS-CoV-2 evolution in chronic infections" for consideration at PLOS Pathogens. As with all papers reviewed by the journal, your manuscript was reviewed by members of the editorial board and by several independent reviewers. In light of the reviews (below this email), we would like to invite the resubmission of a significantly-revised version that takes into account the reviewers' comments.

Both reviewers appreciated the novelty of this work and think that the manuscript presents interesting and exciting results. However, both reviewers indicate that the text that presents the method is not sufficiently clear for readers to understand. The second reviewer further notes that some of the results are counterintuitive. Because of these concerns, we are requesting a major revision of this work that addresses these important issues.

We cannot make any decision about publication until we have seen the revised manuscript and your response to the reviewers' comments. Your revised manuscript is also likely to be sent to reviewers for further evaluation.

Sincerely,

Katia Koelle

Guest Editor

PLOS Pathogens

Ronald Swanstrom

Section Editor

PLOS Pathogens

Michael Malim

Editor-in-Chief

PLOS Pathogens

orcid.org/0000-0002-7699-2064

Reviewer's Responses to Questions

**Part I - Summary**

Reviewer #1: This is a very interesting study investigating the within host evolutionary rate of SARS-CoV-2. The authors developed novel methods to estimate such rates and used a very well curated data set for this purpose. The key findings are that the virus often displays population structure within hosts and that it can evolve much faster than estimates at the host-population level.

Overall, I think that this is a solid piece of work and I do not have fundamental concerns. I would like, however, to point out a few things that I deem necessary to improve the clarity. I anticipate that they will require additional work and rewriting, but not additional analyses:

Reviewer #2: This manuscript analyzes sequencing data from chronic SARS-CoV-2 infections and finds evidence in multiple patients for the co-existence of multiple viral lineages, sometimes evolving at different rates. This is an important topic (as the manuscript points out, the Variants of Concern were likely produced by evolution within chronic infections) and this study advances our understanding by characterizing how the virus diversified in multiple infections. So at a high level I found this to be an exciting manuscript. My main concern is that I couldn't understand the central method of the paper for inferring the subpopulation structure, and in the end I'm not even sure how to interpret a subpopulation in this context.

**Part II – Major Issues: Key Experiments Required for Acceptance**

Reviewer #1: - The methods are not detailed enough to assess them properly. I am particularly interested in the inference of subpopulations, but I had to sketch it by hand to grasp what is actually going on. The equations referring to Poisson processes are useful, yet I would encourage the authors to include something graphic to illustrate the initial consensus sequence and how it can evolve into populations or not. This is an important point because this method could potentially become very useful to other researchers if they can follow it.

- One of the github repositories does not work (the link leads to an error) and the other has no documentation, only C++ code. I do not think that this needs to be very extensive, but sufficient guidelines to analyse a toy data set would be immensely useful.

- There needs to be more detail with the phylogenetic tree inference. Because the data involve very low genetic divergence, then the substitution model and likelihood replicates are especially important to determine the reliability of the inferences. I am sure there are no errors here, a few lines with details are needed (substitution model used, etc…).

Reviewer #2: 1. There were many details in the description of the method that I could not understand, but at an overarching level, I think it would help to have a clearer description of the biology that the method is trying to capture, say a description of the authors' underlying mental model for the population that drove their methodological choices. I think I can mostly guess it from the setup of the simulations, but not entirely. As an example of how such a biological motivation could help, I think I could follow the algorithm in lines 424-435 as just an abstract series of steps. But it would be much easier to grasp if the manuscript told us what the steps meant biologically.

2. I am particularly concerned about the method because it seems to be producing some counterintuitive results. As I understand Figure 5B, it seems to be saying that there is extensive allele sharing across subpopulations, with the major mutant alleles in each subpopulation appearing as minor alleles in the other subpopulations. How are we supposed to interpret this? What do these subpopulations represent? It would seem to be much more natural to think in terms of lineages. Then there would simply be three mutant lineages (D796H / Delta69-70, P330S / W640G, and T240I / Y200H), each present at different frequencies at the different timepoints. The rate of evolution within a lineage is then the rate of substitutions along a line of descent. Since the current subpopulations almost line up with lineages, it seems like this change could be made without much change to the downstream results, but with a huge increase in clarity and interpretability.

**Part III – Minor Issues: Editorial and Data Presentation Modifications**

Reviewer #1: Lines 177 to 179: I found it very interesting that the upper range for the evolutionary rate, of 3.9E-3 subs/site/year, for within host evolution here is comparable to phylogenetic estimates of the precursor lineages of variants of concern (about 2.45E-3 subs/site/year, Tay et al. 2022) using consensus sequences from individual patients. Can the authors comment on whether this may suggest a role in subpopulation structure in the emergence of variants (probably in the Discussion)?

L 244: One reason for why regression based estimates might be lower is the presence of pseudoreplication. Conceptually, if evolution is tree like and we assume some degree of shared ancestry between samples, then the deep branches in a tree would be traversed multiple times. These branches tend to have fewer transitory mutations and because they are counted multiple times, they can impose a bias on estimates of evolutionary rates. This is a hypothesis I have worked with, but would be most interested to know what the authors think.

Reviewer #2: 1. In Fig 3B, the regression slope looks close to 1. Might be worth adding an SI figure showing the non-synonymous rate vs the combined synonymous and non-coding rate.

2. Lines 322-326: These lines should come earlier---I was struggling to understand Figure 5 without them. But I am also troubled by this assumption. I guess this is what is leading to the allele sharing. In the lineage picture, it seems much more natural to just say that multiple lineages are allowed to be present in each sample. The manuscript cites two papers in support of its "either-or" assumption. Both had the senior author of the present manuscript as an author, so I'm a bit hesitant about asserting this, but I believe that they instead show that samples tend to be "dominated" by one lineage, but with other lineages still found at low frequency. Fig 3a of the first (Kemp et al) just shows the same patient as Fgi 5 of the present manuscript, and one can see the minority lineages in the same time points. The second paper is on a quite different virus, influenza B, but still if I'm reading its Fig 1 correctly, on one of the two days (Oct 4) when clade B was observed, clade A was also observed.

3. The description of how the starting genotype is inferred when there is no sequence from the onset of infection should be clarified. Is Wuhan Hu-1 used to polarize the mutations, as in the simulations? Are subpopulations allowed to share substitutions? (Lines 420-421 seem to suggest that they can share one, but the fact that there is no ambiguity in the original genotype for >2 subpopulations suggests that they can't.)

4. I don't understand how not allowing subpopulations to share substitutions is a form of parsimony. Why can't one lineage acquire substitutions and then split into two that acquire different subsequent substitutions? That doesn't require any more events than a case where the splitting occurs before the substitutions.

5. The explanation of the motivation behind the dimensionality reduction approach used in Figure 1 should be clearer. What are we supposed to take away from these plots that would be obscured by, e.g., PCA? Honestly, I think Fig 1 would be stronger if it just showed the trees, with the lineage colors. The scatter plots just confused me and I didn't even notice the trees at first.

6. As another example of how more explanation of the biology behind the method would be helpful, I'm still not sure how to think of the epsilon parameter. Lines 313-315 say the definition "is a broad one, encompassing for example the temporary changes in consensus that might result from clonal competition". But I don't think it's ever explained what the definition actually is. And couldn't the changes in consensus sequence in Fig 5 be considered to result from clonal competition? What does the parameterization of epsilon on lines 464-466 imply about what processes are contributing to it?

PLOS authors have the option to publish the peer review history of their article (what does this mean? ). If published, this will include your full peer review and any attached files.

**Do you want your identity to be public for this peer review?** For information about this choice, including consent withdrawal, please see our Privacy Policy .

Reviewer #1: No

Reviewer #2: No
---

## [Decision Letter · Decision Letter 1]

18 Feb 2025

PPATHOGENS-D-24-01172R1

Variable rates of SARS-CoV-2 evolution in chronic infections

PLOS Pathogens

Dear Dr. Illingworth,

Thank you for submitting your manuscript to PLOS Pathogens. After careful consideration, we feel that it has merit but does not fully meet PLOS Pathogens's publication criteria as it currently stands. Therefore, we invite you to submit a revised version of the manuscript that addresses the points raised during the review process.

Please submit your revised manuscript within 30 days Apr 19 2025 11:59PM. If you will need more time than this to complete your revisions, please reply to this message or contact the journal office at plospathogens@plos.org. Please include the following items when submitting your revised manuscript:

We look forward to receiving your revised manuscript.

Kind regards,

Katia Koelle

Guest Editor

PLOS Pathogens

Ronald Swanstrom

Section Editor

PLOS Pathogens

Sumita Bhaduri-McIntosh

Editor-in-Chief

PLOS Pathogens

orcid.org/0000-0003-2946-9497

Michael Malim

Editor-in-Chief

PLOS Pathogens

orcid.org/0000-0002-7699-2064

**Additional Editor Comments :**

Both reviewers appreciate the substantial revisions that the authors have made in response to the first set of reviews. While the first reviewer is happy with this revised manuscript, the second reviewer has several concerns that have arisen from now having a better understanding of the methodology. The concerns include whether subpopulation structure may be supported when evolutionary rate heterogeneity is the reason for observing a burst of new alleles as well as whether the ambiguous nucleotides in Patient G are meaningful (in that they might indicate something other that low sequencing quality). The reviewer also has additional major concerns. While I am recommending a minor revision, a thorough response to this reviewer's current concerns I think will require substantial thought and potentially several reanalyses, such that edits may end of being closer to a major revision.

**Journal Requirements:**

1) Thank you for uploading your study's underlying data set. We notice that there is a GPL-3.0  license on your data. We would encourage you to consider using a license that is no more restrictive than CC BY, in line with PLOS’ recommendation on licensing (http://journals.plos.org/plosone/s/licenses-and-copyright). 

**Reviewers' Comments:**

Reviewer's Responses to Questions

**Part I - Summary**

Reviewer #1: I would like to thank the authors for revising their manuscript. I believe that the present version is clearer and I would like to highlight the following points:

- Figs and and 7 are very useful for understanding the methods and my earlier point about the inference of subpopulations.

- The github repository is now accessible and it contains enough information to conduct analyses, obtain the data, etc…

- I had not previously understood that this method does not explicitly use phylogenetic trees. This makes sense!

- I appreciate the authors’ response about pseudoreplication. This is an important point when measuring multiple genetic distances and they make a good argument about how it increases the variance in rate estimates, but not necessarily a systematic bias.

- The discussion about the potential role of within host compartments in the emergence of variants, or simply very divergent lineages is an important one to make and which may also to hold in other viruses (e.g. mpox), pending more evidence from deep sequencing, for example. I am not suggesting any edits, but I thought I would mention this possibility.

I have no further comments and did not spot any particular errors or typos in the revised version.

Reviewer #2: The authors have greatly clarified their methods and the underlying data.

Now that I understand things better, I am concerned about the level of support for the subpopulation model and the values of the evolutionary rates in patients E and F. If I understand Fig S1 correctly, in two of these cases, the evidence is simply that there was a burst of multiple substitutions in the last sample after a period of stasis in the first three samples---no parallel diversification or anything like that. I think the model favors subpopulations here because it requires substitution rates to be constant over time, while allowing them to vary across subpopulations. But in Patient H (Figure 5), it looks like in each subpopulation there is a long period of stasis followed by bursts of multiple substitutions, so I don't think this should be regarded as an unlikely scenario. To some extent, this simply means that we could re-interpret the results as being about rate heterogeneity in time rather than space, but it would make the heterogeneity even more extreme.

I am also concerned about how many ambiguous nucleotides there are in the consensensus sequences at variable sites in patient G. Do the authors have any sense for what is going on here? Are these just low-quality sequences with many ambiguous sites throughout the genome, or is there something special about these sites? Perhaps polymorphism is contributing to ambiguity?Either way, it's a bit worrisome as it is, but if it's the latter and really is reflecting polymorphism, it would be a great opportunity to make the analysis much stronger.

**Part II – Major Issues: Key Experiments Required for Acceptance**

Reviewer #1: (No Response)

Reviewer #2: 1. "Fixation and fluctuation sites": I'm still stuck at the point where I was before, being able to follow the algorithm as a pure algorithm (I think), but not understanding the biology. For example, suppose I have an individual sampled 6 times, and I try to write down the maximally fluctuating sequence of observations of a variant, 010101. If I'm understanding the algorithm correctly, this would be classified as a fixation, rather than a fluctuation. (The Hamming distance to both 000000 and 111111 is 3, while the distance to 011111 is 2.) The same would be true for any permutation (necessarily leaving the first 0 in place): 011100, 011010, etc. Similarly, in the real data, variant 6 in subpopulation of Patient G appears in the sequence 01010N1 and is called a fixation. What is the biological interpretation of this?

2. I still don't totally understand how the initial genotype is inferred. In Model X, if there is a polymorphism in which two subpopulations have one allele and two subpopulations have the other, how is it determined what the initial sequence was? In both Model X and Model Y, what happens when sites are polymorphic within a subpopulation and both alleles appear in other subpopulations?

**Part III – Minor Issues: Editorial and Data Presentation Modifications**

Reviewer #1: (No Response)

Reviewer #2: 1. Having the consensus sequence represent a single subpopulation is a very mild assumption. Even without any subpopulation structure, you'd expect that typically the consensus sequence would reflect a single lineage rather than a chimera. I think the authors can be pretty confident here. Maybe Figure 5 could be redone to show how the consensus sequence is always from just one lineage even when other lineages are detected?

2. Line 566, "the assumption that variants may be gained but never lost": I am not sure what this phrase means---columns do go from 1 to 0 within a subpopulation in the data.)

3. Lines 167-169: "We note that the variants C5T and C7T could not be distinguished by our method: Under our maximum likelihood reconstruction one was inferred to be a fluctuation event, with the other being a fixation.": I would take another sentence or two to clarify this.

PLOS authors have the option to publish the peer review history of their article (what does this mean? ). If published, this will include your full peer review and any attached files.

**Do you want your identity to be public for this peer review?** For information about this choice, including consent withdrawal, please see our Privacy Policy .

Reviewer #1: No

Reviewer #2: No

**Figure resubmission:**
---

## [Decision Letter · Decision Letter 2]

8 Apr 2025

Dear Dr Illingworth,

We are pleased to inform you that your manuscript 'Variable rates of SARS-CoV-2 evolution in chronic infections' has been provisionally accepted for publication in PLOS Pathogens.

Best regards,

Katia Koelle

Guest Editor

PLOS Pathogens

Ronald Swanstrom

Section Editor

PLOS Pathogens

Sumita Bhaduri-McIntosh

Editor-in-Chief

PLOS Pathogens

orcid.org/0000-0003-2946-9497

Michael Malim

Editor-in-Chief

PLOS Pathogens

orcid.org/0000-0002-7699-2064

We thank the authors for their thorough response to the reviewers' concerns, which have all been addressed in this second revision.

Reviewer Comments (if any, and for reference):

Reviewer's Responses to Questions

**Part I - Summary**

Reviewer #2: The authors have thoroughly addressed all my concerns. I apologize for not fully understanding the first version of the manuscript, and thank them for their patience.

**Part II – Major Issues: Key Experiments Required for Acceptance**

Reviewer #2: (No Response)

**Part III – Minor Issues: Editorial and Data Presentation Modifications**

Reviewer #2: My previous review's minor comment 1 should have included a line number reference. I was talking about lines 379-381 (in the current version), "Finally, we assumed that each consensus sequence could be assigned uniquely to a single subpopulation." This is a pretty mild assumption because even if a sample contains multiple subpopulations, typically one will be more common than the other in the sample and will match the consensus.

PLOS authors have the option to publish the peer review history of their article (what does this mean? ). If published, this will include your full peer review and any attached files.

**Do you want your identity to be public for this peer review?** For information about this choice, including consent withdrawal, please see our Privacy Policy .

Reviewer #2: No

---

## [Editor Report · Acceptance letter]

Dear Dr Illingworth,

We are delighted to inform you that your manuscript, "Variable rates of SARS-CoV-2 evolution in chronic infections," has been formally accepted for publication in PLOS Pathogens.

Best regards,

Sumita Bhaduri-McIntosh

Editor-in-Chief

PLOS Pathogens

orcid.org/0000-0003-2946-9497

Michael Malim

Editor-in-Chief

PLOS Pathogens

orcid.org/0000-0002-7699-2064